# FROM MNIST TO IMAGENET: UNDERSTANDING THE SCALABILITY BOUNDARIES OF DIFFERENTIABLE LOGIC GATE NETWORKS

## ABSTRACT

Differentiable Logic Gate Networks (DLGNs) are a very fast and energy-efficient alternative to conventional feed-forward networks. With learnable combinations of logical gates, DLGNs enable fast inference by hardware-friendly execution. Since the concept of DLGNs has only recently gained attention, these networks are still in their developmental infancy, including the design and scalability of their output layer. To date, this architecture has primarily been tested on datasets with up to ten classes.

This work examines the behavior of DLGNs on large multi-class datasets. We investigate its general expressiveness, its scalability, and evaluate alternative output strategies. Using both synthetic and real-world datasets, we provide key insights into the importance of temperature tuning and its impact on output layer performance. We evaluate conditions under which the Group-Sum layer performs well and how it can be applied to large-scale classification of up to 2000 classes.

## 1 INTRODUCTION

Deep artificial neural networks have improved immensely in the last few years, exhibiting impressive performance across a wide range of tasks (Golroudbari & Sabour, 2023; Noor & Ige, 2024; Ekundayo & Ezugwu, 2025). However, these improvements come with rapidly growing computational costs (Thompson et al., 2020; Rosenfeld, 2021; Tripp et al., 2024). This constrains their deployment in many real-world environments, particularly on edge devices and mobile phones (Zhang et al., 2020; Zheng, 2025). Thus, there is increasing interest in developing neural networks with competitive performance and energy-efficient deployment.

All computations on digital hardware are inherently built from Boolean operations such as AND, OR, and NOT (Kukunas, 2015). This raises the question of whether machine learning models can be run directly on logic gates, the fundamental building blocks of digital computation.

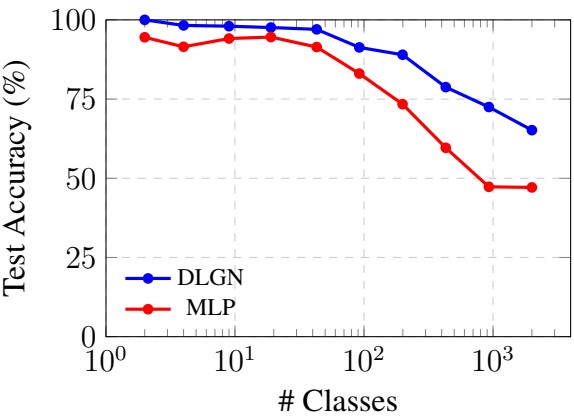

Figure 1: DLGNs (blue) consistently outperform MLPs (red) across classification tasks with up to 2000 classes. The result illustrates the potential of logic-gate-based architectures to remain effective when applied to large-scale classification problems.

Logic Gate Networks (LGNs) provide one way to address this question. Instead of relying on traditional arithmetic operations, LGNs combine discrete logical operations, enabling extremely fast inference. While inference is efficient, training such discrete networks poses significant challenges. Differentiable LGNs (DLGNs) (Petersen et al., 2022) resolve this issue by introducing continuous

relaxations of logical operations, allowing LGNs to be trained with gradient-based optimization methods (LeCun et al., 2015; Goodfellow et al., 2016).

Up to now, DLGNs have been evaluated mainly on small classification datasets. Designing an expressive yet trainable classification layer for DLGNs is not trivial. The most common approach is the Group-Sum layer, where a large set of output neurons represents each class. The activations of neurons within each set are summed to produce the logit for that class. Thus, every class requires its own dedicated group of neurons. Petersen et al. (2022) report using between 8'000 and 64'000 output neurons for MNIST (800–6'400 neurons per class) and up to 102'400 neurons per class for CIFAR-10. While effective for small-scale datasets, this design raises concerns about efficiency and scalability as the number of classes increases.

The standard Group-Sum classification layer is believed to have limited capacity to handle larger numbers of classes, potentially restricting the scalability of DLGNs. Petersen et al. (2022; 2025); Yousefi et al. (2025) mainly evaluated DLGNs on MNIST and CIFAR-10, arguing that training for a larger number of classes is infeasible when up to 102'400 neurons per output class is required.

In this work, we provide the first large-scale evaluation of DLGNs on datasets with thousands of classes, systematically analyzing the expressiveness of the Group-Sum output layer. We show that the temperature parameter $\tau$ is a key factor that controls redundancy and neuron utilization, directly influencing scalability. Beyond Group-Sum, we propose and evaluate alternative output layer designs, comparing their effectiveness across synthetic and real-world datasets. Together, these experiments shed light on the strengths and limitations of DLGNs in large-class settings and highlight open challenges for extending these architectures to more complex data. Open questions include how many output bits are needed to represent a class reliably and whether summing over large groups of output neurons provides an effective decoding strategy.

## 2 BACKGROUND

### 2.1 DIFFERENTIABLE LOGIC GATE NETWORKS

Logic Gate Networks (LGNs) are composed of Boolean logic gates that process binary signals. Karakatic et al. (2013) proposed a genetic programming approach that constructs circuits from truth tables. While effective for small tasks, these methods scale poorly (Ondas et al., 2005). Differentiable Logic Gate Networks (DLGNs) (Petersen et al., 2022) address this limitation by introducing continuous relaxations of discrete functions, enabling gradient-based training.

A DLGN consists of *LogicLayers*, where each neuron receives two inputs and applies a learnable logical function. During training, a neuron's output is computed as:

$$o = \sum_{i=1}^{16} p_i \cdot f_i(a, b) = \sum_{i=1}^{16} \frac{e^{w_i}}{\sum_{j=1}^{16} e^{w_j}} \cdot f_i(a, b), \tag{1}$$

where $a$ and $b$ are inputs, $f_i$ represent logical functions such as AND, OR, XOR (see Table 1), and $w_i$ are learnable weights. The continuous formulation allows end-to-end training with gradient-based learning methods (LeCun et al., 2015; Goodfellow et al., 2016).

During inference, only the function with the largest weight is used:

$$o = f_{i^*}(a, b), \quad i^* = \arg\max_{i \in \{1, \dots, 16\}} w_i. \tag{2}$$

This reduces computation to binary logical operations, enabling highly efficient predictions. This is referred to as the discrete setting. Here, the inputs must also be binarized.

### 2.2 GROUP-SUM LAYER

The Group-Sum layer serves as the DLGN output layer. The output of the final layer ($\mathbf{o}$) is partitioned into $k$ equal segments, one per class. The outputs in each segment are summed and passed through a softmax to form the predicted probability distribution:

$$\mathbf{p} = softmax\left(\frac{1}{\tau} \left[\sum_{j=0}^{\frac{n}{k}-1} o_j, \sum_{j=\frac{n}{k}}^{\frac{2n}{k}-1} o_j, \dots, \sum_{j=\frac{(k-1)n}{k}}^{n-1} o_j\right]\right), \tag{3}$$

Table 1: List of real-valued binary logic operators used in the neurons of a Differentiable Logic Gate Network. During training, the real-valued functions are used to allow gradient propagation, thus enabling gradient-based learning methods (LeCun et al., 2015; Goodfellow et al., 2016).

| ID | Operator | Real-valued equivalent | 00 | 01 | 10 | 11 |
|---|---|---|---|---|---|---|
| 0 | FALSE | $0$ | 0 | 0 | 0 | 0 |
| 1 | $a \wedge b$ | $a \cdot b$ | 0 | 0 | 0 | 1 |
| 2 | $\neg(a \Rightarrow b)$ | $a - ab$ | 0 | 0 | 1 | 0 |
| 3 | $a$ | $a$ | 0 | 0 | 1 | 1 |
| 4 | $\neg(a \Leftarrow b)$ | $b - ab$ | 0 | 1 | 0 | 0 |
| 5 | $b$ | $b$ | 0 | 1 | 0 | 1 |
| 6 | $a \oplus b$ | $a + b - 2ab$ | 0 | 1 | 1 | 0 |
| 7 | $a \vee b$ | $a + b - ab$ | 0 | 1 | 1 | 1 |
| 8 | $\neg(a \vee b)$ | $1 - (a + b - ab)$ | 1 | 0 | 0 | 0 |
| 9 | $\neg(a \oplus b)$ | $1 - (a + b - 2ab)$ | 1 | 0 | 0 | 1 |
| 10 | $\neg b$ | $1 - b$ | 1 | 0 | 1 | 0 |
| 11 | $a \Leftarrow b$ | $1 - b + ab$ | 1 | 0 | 1 | 1 |
| 12 | $\neg a$ | $1 - a$ | 1 | 1 | 0 | 0 |
| 13 | $a \Rightarrow b$ | $1 - a + ab$ | 1 | 1 | 0 | 1 |
| 14 | $\neg(a \wedge b)$ | $1 - ab$ | 1 | 1 | 1 | 0 |
| 15 | TRUE | $1$ | 1 | 1 | 1 | 1 |

where $n$ is the number of output neurons, $k$ the number of classes, and $\tau$ a temperature scaling.

## 2.3 THE ROLE OF $\tau$

Temperature $\tau$ strongly affects performance (see Section 5). Small $\tau$ values produce sharper predictions and larger gradients, increasing confidence but potentially destabilizing training. Large $\tau$ values result in smooth predictions, reducing gradients and model confidence. Section 5.4 provides a detailed analysis.

## 3 RELATED WORK

The development of Differentiable Logic Gate Networks (DLGNs) can be seen as an extension of earlier work in logic-based neural computing (Karakatic et al., 2013). These networks struggle to scale effectively to larger architectures (Karakatic et al., 2013; Ondas et al., 2005).

Differentiable Logic Gate Networks (DLGNs) (Petersen et al., 2022; 2025) overcome this limitation by relaxing discrete logic functions into continuous approximations. This continuous relaxation enables end-to-end training using gradient-based optimization. DLGNs achieve remarkable computational efficiency, processing over one million MNIST images per second on a single CPU core. When implemented on an FPGA, they are even more efficient, consuming very little power. This makes them suitable for battery-powered edge devices.

Extensions of DLGNs have explored different architectural and application domains. Recurrent Deep Differentiable Logic Gate Networks (RDDLGNs) (Bührer et al., 2025) adapt the logic-based framework to sequence-to-sequence tasks such as neural machine translation. They replace standard neural building blocks with logic operations and achieve performance comparable to GRU baselines. Differentiable Logic Gate Cellular Automata (Miotti et al., 2025) apply DLGNs to learn local update rules in discrete state spaces. This reduces computational cost compared to traditional neural cellular automata while preserving the ability to learn rules.

A parallel line of research focuses on low-precision networks for efficient inference on edge devices. Reducing numerical precision from 32-bit floating-point to 8-bit, 4-bit, or even binary representations substantially accelerates computation with minimal accuracy loss (Rehm et al., 2021; Dettmers, 2016; Sun et al., 2019; 2020; Qin et al., 2020). Techniques like Differentiable Soft Quantization (DSQ) (Gong et al., 2019) mitigate the accuracy gap by approximating full-precision behavior dur-

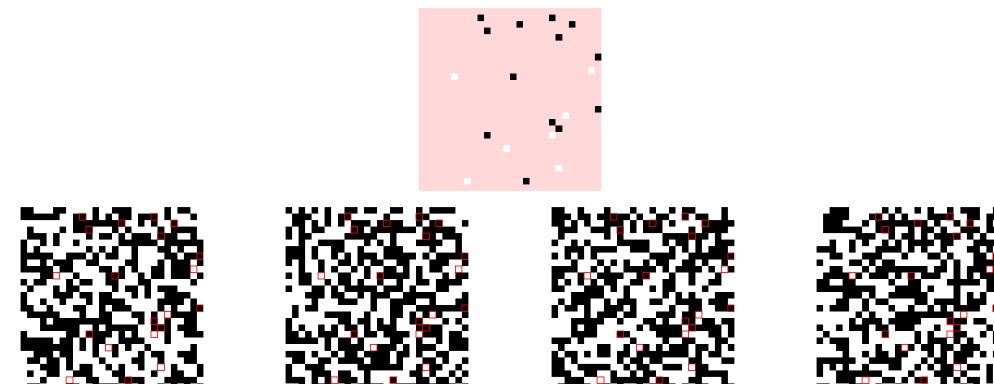

Figure 2: Top row: illustration of class-specific position sampling and initialization in the synthetic dataset. For each class, a random subset of input positions is chosen and fixed to either 0 or 1, defining the class identity. The remaining positions are left unconstrained and are randomly assigned for each individual sample. Bottom row: four complete examples generated for the same class, demonstrating that all samples share the fixed positions while the random positions vary across instances. This design ensures that the dataset is easy to separate at the feature level, so performance differences can be attributed primarily to the capacity of the output layer rather than the backbone.

ing training. These methods share the principle of combining discrete or low-precision operations with gradient-based optimization, conceptually related to DLGNs.

Other work has addressed the discretization gap inherent to differentiable logic networks. Yousefi et al. (2025) introduced Gumbel Logic Gate Networks (GLGNs), injecting Gumbel noise during training to reduce the mismatch between training and inference. This improves neuron utilization and enhances scalability. Gumbel noise has also been shown to act as a regularization technique improving downstream performance (Kim, 2023).

Similarly to DLGNs, differentiable Neural Architecture Search (NAS) methods such as DARTS (Liu et al., 2018) leverage continuous relaxation of discrete design choices to automate the search for high-performing architectures. These methods illustrate a broader trend of using continuous approximations to enable efficient optimization in discrete or combinatorial domains (Zoph & Le, 2017; Dong & Yang, 2019; Baymurzina et al., 2022).

Despite these advances, DLGNs have been evaluated mainly on small-scale classification tasks with up to 10 classes. The standard Group-Sum classification layer, which represents each class with large groups of output neurons, may not scale efficiently to problems with many classes. Our work addresses this gap by investigating the expressiveness and scalability of the DLGN output layer.

## 4 Methodology and Experimental Setup

### 4.1 Datasets

We evaluate how the performance of DLGN models scales across datasets. Specifically, we construct a synthetic dataset, we use the ImageNet-32 dataset, and we combine multiple MNIST variants.

We first introduce a synthetic dataset designed to support dynamically increasing class counts. The dataset is intentionally simple to ensure that the feature extractor can learn effectively, so that any limitations in performance can be attributed to the Group-Sum layer rather than an insufficient feature extraction. Each sample is represented as a binary vector of length 784, matching the dimensionality of MNIST-like datasets Lecun et al. (1998). For each class, between 5 and 40 input positions are randomly chosen and fixed to either 0 or 1, while the remaining positions are assigned randomly for each sample. Figure 2 shows an example of a class and four samples drawn from the class. The top image shows the random sampling of positions and their initialization. All samples of a class share these positions and values. All other values are chosen randomly per sample (four samples are shown on the bottom row).

We evaluate DLGNs on an RGB dataset of higher complexity than the previously evaluated CIFAR-10, namely on the ImageNet-32 dataset that consists of the ImageNet images that have been down-scaled to 32 by 32 (Krizhevsky et al., 2012; Russakovsky et al., 2015; Chrabaszcz et al., 2017). ImageNet-32 scales up to 1,000 classes, making it a particularly challenging benchmark for large-scale classification. DLGNs have not been scaled to larger images (in resolution) than the 28 by 28 for CIFAR images. Therefore, we focus on ImageNet-32 as they are roughly of the same size. DLGNs take a long time to train, so scaling them to larger resolutions is difficult and outside the scope of this work (Petersen et al., 2025; Yousefi et al., 2025; Bührer et al., 2025).

DLGNs perform best on binarized grayscale images, so we construct a dataset with many classes by combining several MNIST-like datasets. These datasets include MNIST Lecun et al. (1998), Fashion-MNIST Xiao et al. (2017), Kuzushiji-MNIST (K-MNIST) Clanuwat et al. (2018), and Q-MNIST Yadav & Bottou (2019).

## 4.2 Input Transformation and Preprocessing

For all MNIST variants, models are trained using continuous inputs without transformation. Preliminary experiments showed negligible performance differences between continuous and binarized inputs for training. A validation set is created by sampling 20% of the training data before training. Unless stated otherwise, references to the validation or test set refer to the binarized version. Binarization is applied by thresholding input values at 0.5.

For CIFAR-10, CIFAR-100, and ImageNet-32, inputs are flattened into vectors of size $32 \cdot 32 \cdot 3 = 3072$ with RGB channels (Petersen et al., 2022). Each vector is expanded using three thresholds, yielding a representation of size $3 \cdot 3072 = 9216$. Formally, an input $x$ is transformed as:

$$f(x) = \text{concat}\left(\text{float}(x > \frac{1}{4}), \text{float}(x > \frac{2}{4}), \text{float}(x > \frac{3}{4})\right). \tag{4}$$

The synthetic dataset requires no transformation, as it is generated directly in binary form. Further details on datasets and preprocessing are provided in Appendix B.

## 4.3 Model Architecture and Training Setup

The DLGN baseline consists of 6 logical layers with 64,000 neurons per layer. The input to the Group-Sum layer, therefore, also counts 64'000 neurons. The 64'000 neurons are then split evenly amongst the classes. As a comparison, we use multilayer perceptrons (MLPs) with three fully connected hidden layers of 256, 512, and 1024 neurons, referred to as *small*, *medium*, and *big*, respectively. A detailed overview of training and architecture parameters, along with complete DLGN and MLP results, is provided in the Appendix B.2 and F. During inference, the models are discretized as described in Section 2.1. This is the default evaluation setting.

## 4.4 Convolutional Logic Gate Networks

As a supplementary evaluation, we also experiment with Convolutional Differentiable Logic Gate Networks (CLGNs) (Petersen et al., 2025). For technical specifications, we refer to the original work. We adopt minimally modified versions of the *M* model for MNIST and CIFAR experiments and the larger *G* model for ImageNet-32, following the configurations in Petersen et al. (2025).

For ImageNet, CIFAR, and the MNIST-like datasets, the input transformations are identical to those in Section 4.2. The use of CLGNs allows us to assess whether conclusions drawn for DLGNs extend to architectures with convolutional backbones. This tests the robustness of our findings across different network families.

## 5 Results

We examine how model performance is affected by increasing the number of classes across three settings: the synthetic dataset, ImageNet-32, and a combined MNIST-like dataset. In addition, we highlight the role of the temperature parameter $\tau$ in enabling the models to scale effectively to a large number of classes.

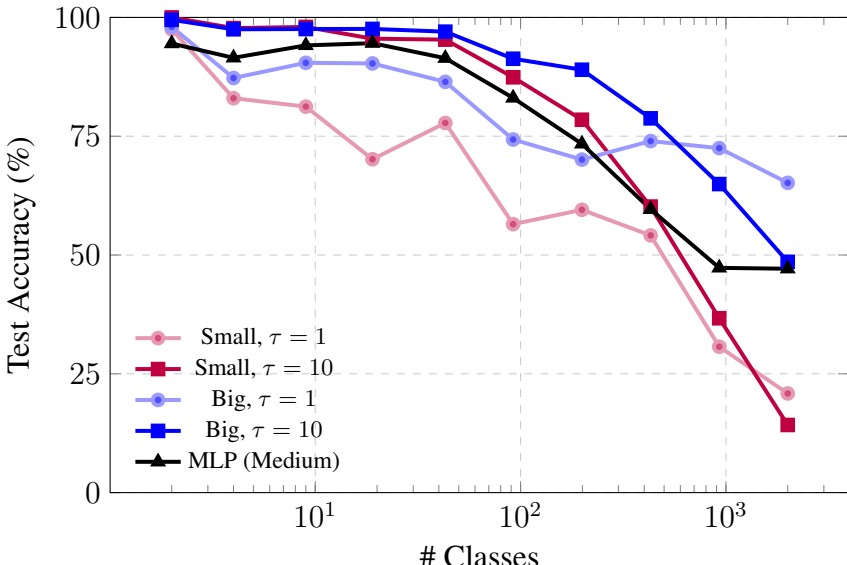

Figure 3: Accuracy of DLGNs compared to the MLP model, considering an increasing number of classes. Small: A DLGN with a layer size of 64'000 logical gates. Big: A DLGN with a layer size of 256'000 logical gates. The MLP model refers to a conventional MLP with three layers of 512 neurons and Batchnorm. The accuracy of all DLGNs stays high up until a few hundred classes, but sharply drops after.

## 5.1 SYNTHETIC DATASET

To evaluate performance with hundreds to thousands of classes while keeping the input size manageable, we construct a synthetic dataset as described in Section 4. The dataset is intentionally simple, ensuring that classification performance is primarily limited by the Group-Sum layer rather than the backbone. For each class, between 5 and 40 input bits are fixed, while the remaining bits are assigned randomly. We scale the number of classes logarithmically from 2 to 2000 and compare four DLGN variants against a medium-sized multilayer perceptron (MLP) baseline. Figure 3 reports accuracy as a function of the number of classes.

The MLP maintains accuracy above 86% up to 100 classes but drops to around 50% at 1000–2000 classes. For DLGNs, performance depends strongly on the choice of $\tau$. When the number of classes is small, each class is represented by a large set of output neurons, and large differences in the summed activations can lead to overconfident predictions. In this regime, higher $\tau$ values are effective, as they temper these differences and prevent a few neurons from dominating the softmax. As the number of classes grows, each class is represented by fewer neurons, reducing the risk of such dominance. Here, smaller $\tau$ values become more suitable, ensuring that the reduced class sums still produce confident and accurate predictions. This trade-off enables DLGNs to remain competitive with the MLP even as the task scales to hundreds or thousands of classes.

Expanding the DLGN backbone to 256'000 neurons per layer (with the same output dimension) yields further gains. With $\tau = 10$, the large DLGN outperforms all models up to 300 classes and continues to exceed the MLP even at 2000 classes. Interestingly, the small DLGN with $\tau = 1$ underperforms on tasks with few classes but performs better as the number of classes increases (see Appendix for additional findings). These results demonstrate that DLGNs can surpass conventional feed-forward networks on large-class problems. Increasing backbone capacity consistently improves performance even when the output dimension is fixed, as representing each class with 32 output neurons in the Group-Sum layer is sufficient to outperform the MLP.

Finally, we investigated the impact of output dimensionality while keeping a 6-layer, 64'000-neuron DLGN backbone fixed. We tested three output sizes (16'000, 64'000, and 256'000 neurons) for $\tau \in \{1, 10, 100\}$. Performance was largely insensitive to output dimension: neither increasing nor decreasing output size had a significant effect on accuracy (see Appendix D.1).

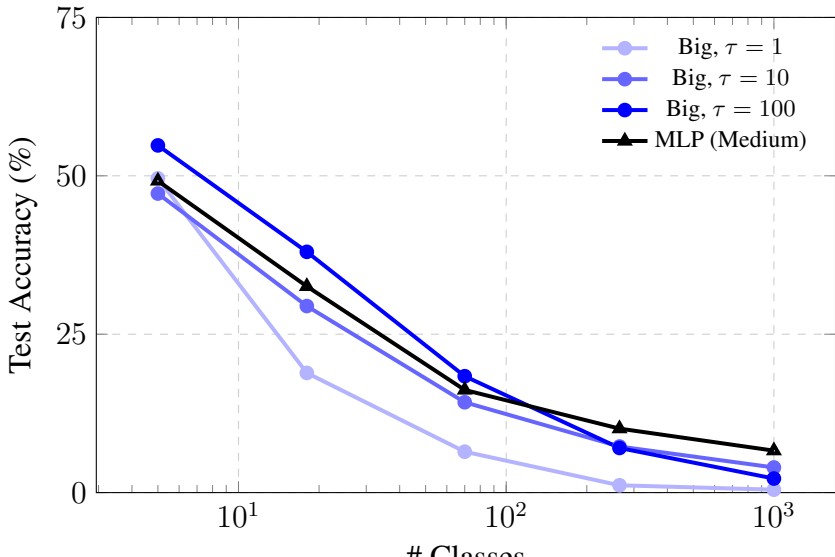

Figure 4: Accuracy of DLGNs compared to the MLP model, considering an increasing number of ImageNet classes. Big: A DLGN with a layer size of 256'000 logical gates. The MLP model refers to a conventional MLP with three layers of 512 neurons and Batchnorm.

## 5.2 IMAGENET-32

To evaluate the applicability of our findings to real-world tasks, we test on the ImageNet-32 dataset. Due to its increased complexity relative to the synthetic dataset, we adopt a DLGN with 256,000 logical gates per layer as our default model and compare it to the same MLP used previously. Both models are trained on binary input representations to avoid any inherent advantage. Performance trends, illustrated in Figure 4, are broadly consistent with those observed on the synthetic dataset: larger $\tau$ values perform better for small numbers of classes, whereas $\tau = 10$ is more effective as the number of classes increases. While $\tau = 100$ allows the DLGN to approach MLP performance for up to 100 classes, no tested configuration matches the MLP beyond that point.

Increasing the DLGN layer size to 512'000 gates (with 512'000 output neurons) does not significantly improve performance.

The discrepancy between synthetic and ImageNet-32 datasets likely stems from several factors. First, the synthetic dataset has a simple, linearly separable structure: certain input features are fixed for specific classes, while the remaining inputs are random. In contrast, ImageNet-32 has higher in-class variability and a complex, noisy input distribution. Second, the input dimension after three thresholds is 9'216. While the MLP effectively has a receptive field covering 100% of the inputs, each DLGN output neuron depends only on $2^n$ inputs across $n$ layers. With six layers, this corresponds to 64 inputs ($\sim 0.7\%$ of the whole input vector), likely insufficient for accurate predictions. For the synthetic dataset, 784 input dimensions result in a larger effective receptive field ($\sim 8\%$). Increasing the number of layers could expand the receptive field but introduces challenges such as vanishing gradients (Petersen et al., 2022).

DLGNs do not perform fundamentally worse than MLPs, as both architectures exhibit decreasing accuracy with an increasing number of classes. Crucially, the best DLGNs achieve performance comparable to MLPs when the temperature parameter $\tau$ is chosen appropriately. In particular, DLGN ($\tau = 10$) achieves results on par with MLPs, indicating that with a well-optimized $\tau$, DLGNs can maintain competitive performance for datasets with up to 67 classes.

## 5.3 CUSTOM MNIST DATASET

DLGN performance is comparable to that of MLPs across different MNIST datasets, likely because grayscale images are relatively easy to classify even in binary form, unlike more complex RGB

Table 2: Model performance on combined MNIST datasets with increasing number of classes. All models are evaluated on binary input representations. The best model per column is shown in **bold**, and the second-best is underlined.

| Model | 2 Classes | 4 Classes | 11 Classes | 27 Classes | 67 Classes |
|---|---|---|---|---|---|
| **MLP (Medium)** | **99.58** | 99.09 | 96.20 | 92.37 | 83.53 |
| **DLGN** ($\tau = 1$) | 97.92 | 97.27 | 91.27 | 73.33 | 56.78 |
| **DLGN** ($\tau = 3$) | 98.65 | 97.69 | 94.05 | 88.92 | 75.44 |
| **DLGN** ($\tau = 10$) | 98.96 | 98.56 | 96.98 | 93.15 | 83.42 |
| **DLGN** ($\tau = 30$) | 99.17 | 98.90 | 97.40 | 93.30 | 78.40 |
| **DLGN** ($\tau = 100$) | 99.27 | **99.20** | 96.87 | 86.32 | 62.12 |
| **CLGN** ($\tau = 1$) | 31.50 | 39.09 | 73.34 | 81.68 | 61.71 |
| **CLGN** ($\tau = 3$) | 31.50 | 16.09 | 39.14 | 89.51 | 85.52 |
| **CLGN** ($\tau = 10$) | 32.25 | 46.05 | 81.34 | 96.56 | **88.29** |
| **CLGN** ($\tau = 30$) | 77.88 | 86.05 | 97.16 | **96.89** | 88.13 |
| **CLGN** ($\tau = 100$) | 97.00 | 98.23 | **98.04** | 96.19 | 80.49 |

datasets such as CIFAR 10 or ImageNet 32. To study scalability, we first combine multiple MNIST datasets, including E-MNIST Balanced, K-MNIST, and Fashion-MNIST, into a single dataset with 67 classes and gradually increase the number of classes in a logarithmic fashion. Test accuracies for the different models, evaluated on binary input representations, are summarized in Table 2.

We evaluated our findings on convolutional differentiable logic gate networks (CLGNs) (Petersen et al., 2025), which generally show similar behavior to feed-forward DLGNs. On all datasets, performance varies a lot with the temperature parameter $\tau$. Combining MNIST-like datasets into a 67-class dataset demonstrates that optimal $\tau$ values are even more important than for DLGNs (see Figure 2). Similarly, smaller $\tau$ are superior for large-class datasets, whereas larger $\tau$ perform better on datasets with small number of classes. This effect can be reduced by using alternative output layer architectures, such as the Codebook-Output layer (see Appendix E).

On ImageNet-32, similarly to DLGN, increasing backbone size improves accuracy, but enlarging the Group-Sum output layer has minimal effect. Additional results are provided in the Appendix.

## 5.4 EFFECT OF $\tau$ ON MNIST DATASETS

We begin by examining the effect of different $\tau$ values on validation accuracy for the MNIST digits dataset, observing similar trends across other datasets. Higher $\tau$ values yield smoother learning curves and faster convergence, while excessively large values (e.g., $\tau = 200$) reduce final accuracy and very small values (e.g., $\tau = 1$) prevent convergence. For MNIST digits, $\tau = 20$ achieves the best performance. Optimal $\tau$ values for other datasets are reported in Appendix F.

The temperature parameter $\tau$ therefore plays a critical role in model performance and should be treated as a primary optimization target.

Next, we examine individual neuron contributions in the output layer. Figure 5 shows neuron activation distributions for $\tau = 1$ and $\tau = 100$. The activation rate of a neuron is defined as the fraction of inputs producing an output of 1.

Low $\tau$ (e.g., $\tau = 1$) produces a pronounced spike in the activation distribution around 0%, 50%, and 100%, indicating many neurons are either consistently inactive ('dead'), fully active ('saturated'), or toggle in a synchronized manner. This results in increased redundancy and less differentiated contributions from individual neurons. In contrast, high $\tau$ (e.g., $\tau = 100$) generates a broader and smoother activation distribution, with neurons exhibiting more varied activity levels. This greater differentiation enhances the network's ensemble-like behavior and supports effective pruning of redundant neurons (details in Appendix D).

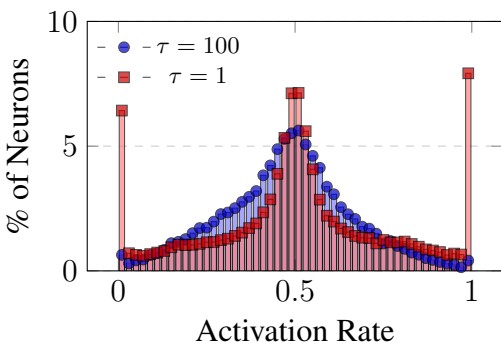
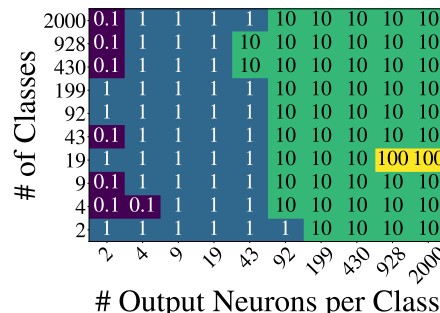

Figure 5: Left: Distribution of neuron activation rates for two models. Larger $\tau$ values concentrate neurons at low activation rates, while smaller $\tau$ shifts the distribution toward higher activations. Right: Best $\tau \in \{0.1, 1, 10, 100\}$ for various numbers of output neurons and neurons per class.

## 5.5 Relationship between Number of Outputs and Temperatur $\tau$

As a supplementary experiment, we want to find the relationship between the output dimension and an optimal $\tau$. We use out synthetic dataset to train models with different output layer size and different number of classes. We chose four different values $\tau \in \{0.1, 1, 10, 100\}$. Figure 5 shows the best $\tau$ for a specific number of classes and number of output neurons per class. Our findings indicate that optimal $\tau$ is not actually dependent on the number of output neurons, but rather on the number of output neurons per class.

## 5.6 Alternatives

To evaluate the effectiveness of the Group-Sum layer, we tested several alternative output layer variants. This analysis identifies the strengths and limitations of the current approach. Some alternatives occasionally approach or slightly surpass the Group-Sum's performance, but none consistently or significantly improve results across datasets. See Appendices E and F for more details and results.

## 6 Conclusion

This work studies the expressiveness and scalability of the Group-Sum output layer in Differentiable Logic Gate Networks. DLGNs have previously been evaluated mainly on datasets with up to ten classes. We extend this analysis to tasks with up to 2000 classes to assess the Group-Sum layer on large-scale classification. Through extensive experiments, we analyze the output layer under different conditions and datasets. We show that the temperature parameter $\tau$ is critical for performance. It affects prediction accuracy, output neuron redundancy, and scalability. We also observe that the optimal value of $\tau$ decreases as the number of output neurons per class increases.

Our results show that DLGNs perform competitively on structured datasets. On MNIST and its variants, DLGNs with the Group-Sum layer achieve accuracy comparable to conventional feedforward networks using binary input data. With a well-chosen $\tau$ parameter, DLGNs maintain high accuracy even with up to 67 classes. On a synthetic dataset, we scale the number of classes up to 2000. In this setting, DLGNs clearly outperform feed-forward networks, demonstrating their ability to distinguish thousands of classes effectively.

Evaluation on the real-world ImageNet-32 dataset highlights current limitations. DLGNs do not achieve performance comparable to feed-forward networks. The complexity of the RGB input and high in-class variability appear to be too challenging for the current network and input representation. This indicates that further architectural adjustments are needed for DLGNs to generalize to natural image datasets.

In conclusion, the Group-Sum output layer is expressive and scalable for structured classification tasks. The choice of $\tau$ is key to achieving high performance. At the same time, DLGNs require further development to improve robustness and generalization on complex real-world data.

## REPRODUCIBILITY STATEMENT

All code used in our experiments is included in the supplementary material, along with a README describing how to run the training and evaluation scripts. The training and test data are publicly available through PyTorch's torchtext, Kaggle, and Huggingface. The code will be made publicly available on GitHub with the camera-ready version. Details of model architectures, training procedures, and datasets are provided in Section 4 and Appendix B.

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

Table 3: Overview of the datasets used in this study.

| Dataset Name | # Samples | Input Dimensions | # Classes |
|---|---|---|---|
| MNIST | 70,000 | $28 \times 28$ (grayscale) | 10 |
| Fashion-MNIST | 70,000 | $28 \times 28$ (grayscale) | 10 |
| Kuzushiji-MNIST | 70,000 | $28 \times 28$ (grayscale) | 10 |
| Q-MNIST | 120,000 | $28 \times 28$ (grayscale) | 10 |
| E-MNIST (Balanced) | 131,600 | $28 \times 28$ (grayscale) | 47 |
| E-MNIST (Letters) | 145,600 | $28 \times 28$ (grayscale) | 26 |
| CIFAR-10 | 60,000 | $32 \times 32$ (RGB) | 10 |
| CIFAR-100 | 60,000 | $32 \times 32$ (RGB) | 100 |
| ImageNet-32 | 1,331,167 | $32 \times 32$ (RGB) | 1,000 |
| Synthetic | 600/Class | 784 (binary) | $2 - 2,000$ |

## A    USAGE OF LLMS

We have made use of several large language models (LLMs) during the preparation of this work. ChatGPT, Claude, Gemini, and Grammarly were employed to assist with spellchecking, improving wording, and shortening text for clarity and readability. In addition, ChatGPT, Claude, and Cursor were used for analyzing and explaining code, providing code completions, and generating visualizations to support our implementation and experiments. These tools were applied as auxiliary aids to polish the writing and streamline the development process, while the core research contributions, experimental design, and interpretation of results remain entirely our own.

## B    IMPLEMENTATION DETAILS

This section details the experimental setup, including descriptions of the datasets used, input transformations applied and the evaluation metrics examined.

### B.1    DATASETS

Table 3 summarizes the various datasets used, their number of samples, input dimensions, and number of classes.

### B.2    TRAINING AND ARCHITECTURAL DETAILS

The most important training and architectural parameters of the baseline DLGN and MLP are presented in Table 4. Additional experiments most often use a minimally modified version of this baseline configuration.

### B.3    EVALUATION METRIC

The model performance is evaluated using **classification accuracy**, which reflects the proportion of correctly predicted samples relative to the total number of samples.

Accuracy is selected as the primary metric because it provides a clear and intuitive measure of overall model effectiveness. It enables straightforward comparisons between different architectures and training configurations. Our dataset exhibits a reasonably balanced class distribution, accuracy therefore serves as a reliable performance indicator.

## C    IMPACT OF $\tau$

Figure 6 shows the impact of $\tau$ on performance for different datasets. Even though the same model is used, the optimal value of $\tau$ greatly differs. While high values perform well on CIFAR-10 dataset (top left), they do not perform nearly as good on CIFAR-100 (top right).

Table 4: Default configurations for DLGN and MLP models, specifically used for creating the baselines. Three different MLPs were tested, referred to as *small*, *medium*, and *big*, with 256, 512, and 1024 neurons per layer, respectively.

| Parameter | DLGN | MLP |
|---|---|---|
| Number of Layers | 6 | 3 |
| Neurons per Layer | 64,000 | 256 / 512 / 1024 |
| Data Augmentation | None | None |
| Dropout | None | None |
| Batch Normalization | – | Enabled |
| Temperature Parameter ($\tau$) | 10 | – |
| Learning Rate | 0.01 | 0.00001 |
| Training Epochs | 100 | 100 |
| Optimizer | Adam | Adam |
| Loss Function | Cross-Entropy | Cross-Entropy |
| Number of Independent Runs | 3 | 3 |

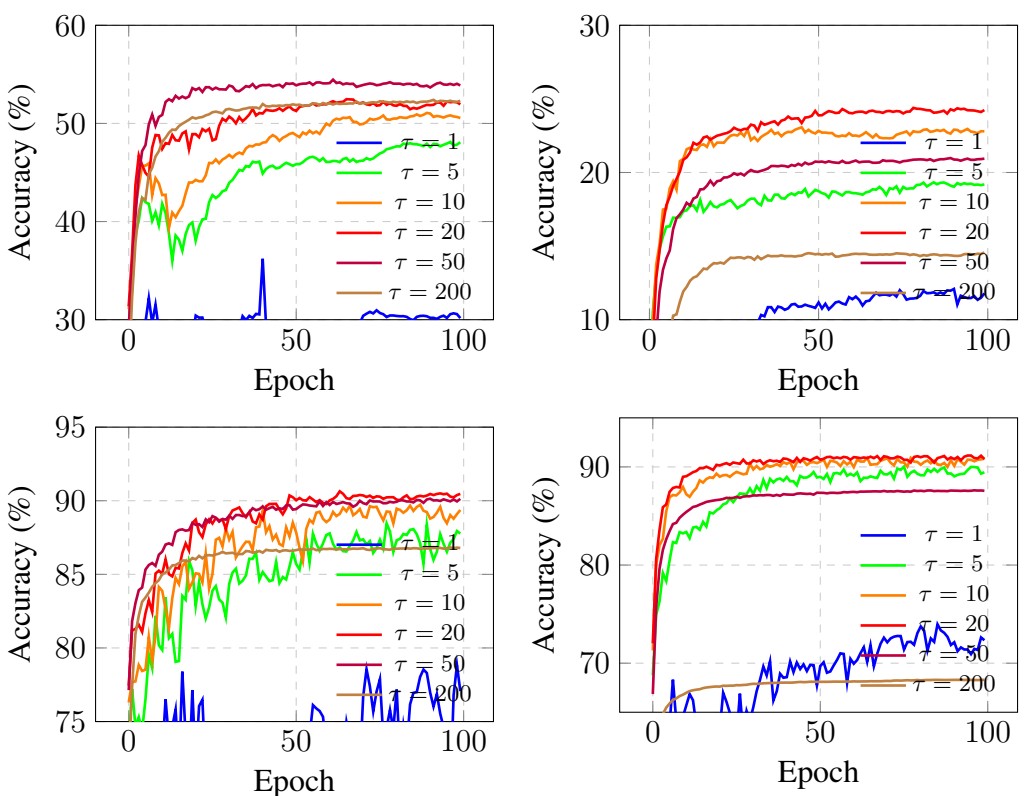

Figure 6: The impact of the $\tau$ value on performance on different datasets. From left to right and top to bottom: CIFAR-10, CIFAR-100, Fashion-MNIST, EMNIST-Letters. The plots show the validation accuracy for different $\tau$. Even though the same model is used, there is great difference between optimal $\tau$.

As mentioned in Section 5.4, per-class accuracy differs greatly for different $\tau$. We analyze this for MNIST. The accuracies are shown in Figure 7. Easily classifiable digits like 1 and 0 show good performance for all $\tau$. However, performance varies a lot more for small $\tau$ as opposed to large $\tau$.

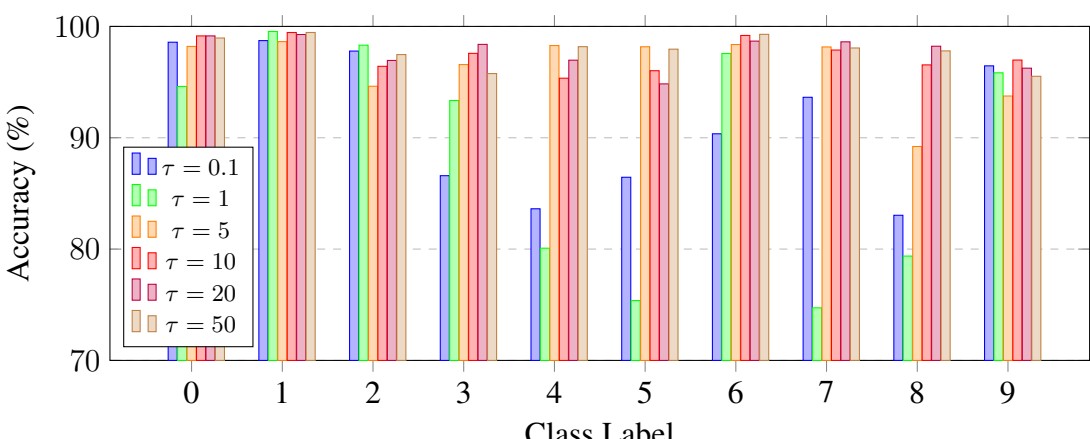

Figure 7: MNIST digits per-class distribution. Low $\tau$ show much greater performance differences compared to large $\tau$.

## D    USING INFORMATION DISTRIBUTION TO PRUNE THE NETWORK

As shown in Section 5, using a large tau value allows us to evenly distribute information across the output neurons. This theoretically allows us to prune some of the output neurons, without loosing much accuracy. We therefore disregard random output neurons for each class. Deleting specific outputs further allows us to prune neurons from intermediate layer. We prune the network by randomly removing output layer neurons. The data is shown in Figure 8. We plot the number of neurons pruned per class to the accuracy that is still preserved. Additionally, in red we show the amount of neurons (in %) that can be pruned in the whole network.

For low tau values, the accuracy curve shows rough characteristics. This indicates that the prediction accuracy of the model heavily depends on specific neurons, rather than on an ensemble of all outputs. The larger $\tau$, the smoother the curve. The information is more evenly distributed over the output neurons, making it possible to remove more neurons, without significant accuracy loss.

### D.1    VARYING OUTPUT LAYER SIZES

In Section 5, we illustrated the performance improvement of DLGNs on the synthetic dataset when increasing the backbone from 64,000 to 256,000 neurons per layer, while keeping the output layer fixed at 64,000 neurons. Here, we further investigate how performance changes when varying the output layer dimension instead. With the backbone fixed at 64,000 neurons per layer, we evaluate models with output layer sizes of 16,000 and 256,000 neurons. Figure 9 compares the results, showing no significant performance differences among the three models. This suggests that accuracy is limited more by the backbone than by the output layer capacity.

## E    ALTERNATIVES

To evaluate the effectiveness of the Group-Sum layer, we investigate several alternative output layer designs, aiming to identify either comparable or superior approaches. Unless stated otherwise, all comparisons are based on a slightly modified version of the standard DLGN baseline described in Section 4. Appendix F contains tables with extensive results for the methods.

### E.1    BINARY LOSS

Instead of using cross-entropy loss, we use a binary logit loss. This loss gets calculated per output neuron instead of over all neurons of a class. Even though the network learns well, its performance does not come close to our baseline.

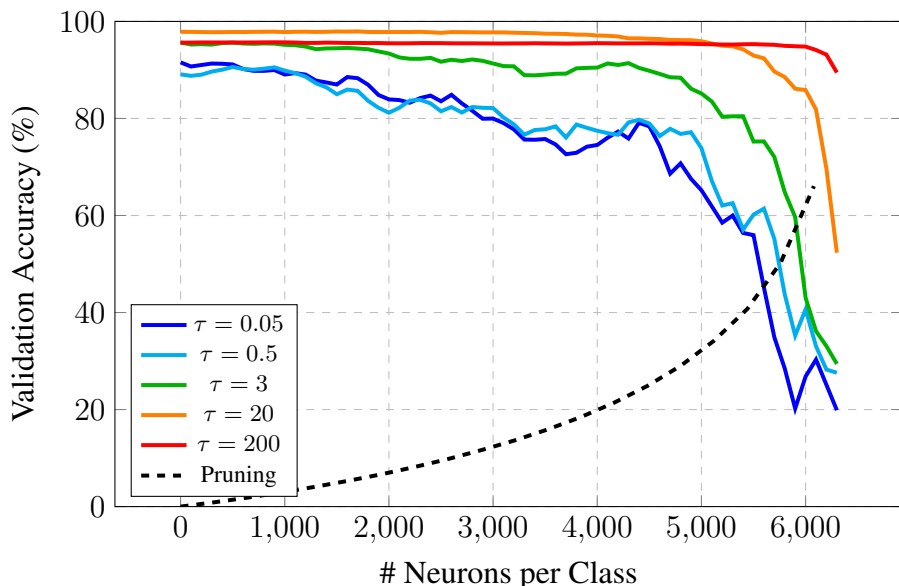

Figure 8: MNIST digits output neurons pruning. Shows the number of output neurons pruned per class (x) to the remaining prediction accuracy. Red indicated the amount of neurons that can be pruned in the whole network.

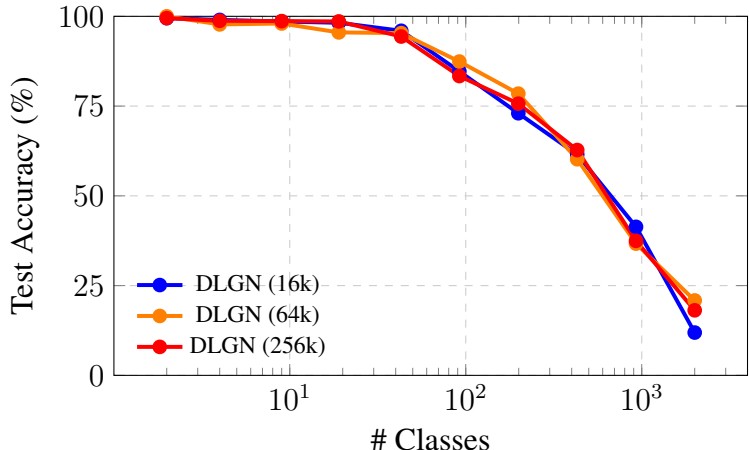

Figure 9: Accuracy of DLGNs for different output layer sizes over increasing numbers of classes, keeping the backbone of the netowork at 64'000 neurons per layer.

### E.2 FULLY-CONNECTED LAST LAYER

To try and get more out of the output layer, we replace the Group-Sum layer with a fully connected layer at the end of the DLGN. This setup could show more of what the network's backbone is capable of. While the performance improves on some datasets, training becomes more unstable.

### E.3 FULLY-CONNECTED AFTER TRAINING

Rather than appending a fully-connected layer during training, we now retrain a separate layer after training the DLGN normally. This increases performance slightly on some of the MNIST datasets.

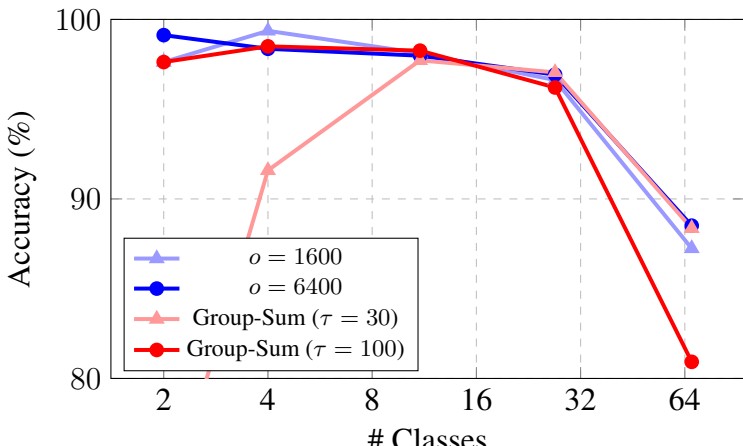

Figure 10: Test Accuracy (eval mode) for MNIST with increasing number of classes. Results compare different output sizes $o$ and Group-Sum with varying $\tau$.

### E.4    CODEBOOK-BASED PREDICTION

Instead of splitting the output layer into $k$ parts for $k$ classes, we try a different method. Each class is assigned a random binary code (a vector of the same length as the output). We then use Hamming distance to compare the network output to each code, picking the closest match. This approach performs better on some datasets — for example, on CIFAR-10, accuracy increases from 50.7% (DLGN baseline) to around 54.8%. Additionally, even though a $\tau$ value is used, the performance is not as dependent on its optimality than with the normal Group-Sum output layer.

We also combine the Group-Sum approach with the Codebook-based approach. By specifying an output size, one can use Group-Sum to create an output with this dimension. We then use the smaller output vector as the network's prediction, calculating the hamming distance to the class encodings. Figure 10 shows the performance of CLGNs on the combined MNIST dataset. In this case we use $\tau = 0.1$. $o$ is the output dimension, equivalent to the dimension of the class encodings. Using the Codebook-Layer and an output dimension of 1600 or 6400, we are able to use a single model to compete or outperform the best Group-Sum layer models for all number of classes.

### E.5    GROUP-SUM DROPOUT

Many of our experimental models show overfitting tendencies. Even though the choice of $\tau$ can help to partially mitigate it, there are other techniques that may be applied. One of them is dropout. Since we are mainly focused on the output layer, we apply dropout only to the Group-Sum layer. We do this by deactivating each neuron with a probability $p \in \{0.1, 0.3, 0.5, 0.7, 0.9\}$ in each batch. Figure 11 shows the performance heatmap of various $\tau$-dropout combinations on our synthetic dataset with 2000 different classes.

We see that certain amounts of dropout increased performance considerably. For $p = 0.1$, almost all test accuracies are superior to $p = 0$. Not only that, but it seems to lessen the importance of an optimal $\tau$, expanding high performance regions.

### E.6    TREE-BASED PREDICTION

Instead of summing up each part of each class, we try to decrease the parts' size by using class-specific DLGN. This halves the parts' dimension with each layer for a certain number of layers before being summed up. We also use a class-specific DLGN that uses the whole output layer for each output. Unfortunately, neither variants improve the model's performance. Additionally, using the whole output layer for each class quickly becomes computationally infeasible for many classes and a large output layer dimension.

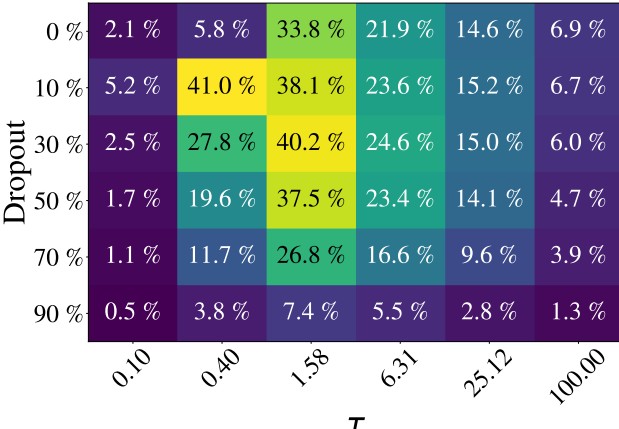

Figure 11: Test performance heatmap of different $\tau$-dropout combinations. The performance is measured on our synthetic dataset with 2000 different classes.

## F    NUMERIC RESULTS

This section displays the resulting accuracies for many of our tested models, including the methods in Appendix E. We show accuracy for all MNIST datasets (Tables 5 to 10) as well as CIFAR10 (Table 11) and CIFAR100 (Table 12). All models were trained with the same input representation as explained in Section 4. DLGNs can only make use of hardware-accelerated inference with binary input representation. To mitigate comparisons of models with inconsistent amounts of input information, we show both the accuracy with binary and continuous input representations. The discrete settings use binary inputs and discrete models for logic gate-based models. *Fully Connected Last Layer* refers to a standard DLGN, with appended fully-connected layers during training. **Run** 1, 2, and 3 refer to appended layers with {512}, {512, 512}, and {1000, 100}, respectively.

Table 5: Test Accuracy (%) Comparison of Models on MNIST (digits).

|  | Input | |
|  | **Discrete** | **Continuous** |
| --- | --- | --- |
| **Baseline** | | |
|  | $97.84 \pm 0.07$ | $98.26 \pm 0.02$ |
| **MLP Baseline** | | |
| Small | $96.81 \pm 0.16$ | $97.83 \pm 0.13$ |
| Medium | $97.19 \pm 0.11$ | $98.12 \pm 0.05$ |
| Big | $97.44 \pm 0.14$ | $98.27 \pm 0.05$ |
| **Different tau** | | |
| $\tau = 1$ | 92.21 | 94.88 |
| $\tau = 3$ | 94.72 | 96.11 |
| $\tau = 10$ | 97.90 | 98.29 |
| $\tau = 30$ | 98.31 | 98.54 |
| $\tau = 100$ | 97.68 | 97.72 |
| **Binary Logit Loss** | | |
|  | 75.68 | 75.35 |
| **Fully Connected Last Layer** | | |
| Run 1 | 95.71 | 97.48 |
| Run 2 | 94.98 | 97.17 |
| Run 3 | 95.68 | 97.16 |
| **Codebook-Based Prediction** | | |
| $\tau = 0.1$ | 97.58 | 98.09 |
| $\tau = 0.3$ | 98.07 | 98.47 |
| $\tau = 1$ | 97.83 | 98.02 |
| $\tau = 3$ | 94.95 | 95.12 |
| **Group-Sum Dropout** | | |
| $p = 0.1$ | 97.72 | 98.12 |
| $p = 0.3$ | 98.15 | 98.30 |
| $p = 0.5$ | 98.16 | 98.38 |
| $p = 0.7$ | 98.24 | 98.29 |
| $p = 0.9$ | 97.28 | 97.30 |
| **Convolutional Difflogic** | | |
| $\tau = 1$ | 92.19 | – |
| $\tau = 3$ | 91.74 | – |
| $\tau = 10$ | 97.14 | – |
| $\tau = 30$ | 98.73 | – |
| $\tau = 100$ | 99.03 | – |

Table 6: Test Accuracy (%) Comparison of Models on Fashion-MNIST.

|  | Input | |
|  | **Discrete** | **Continuous** |
| --- | --- | --- |
| **Baseline** | | |
|  | $78.35 \pm 0.11$ | $88.90 \pm 0.13$ |
| **MLP Baseline** | | |
| Small | $78.90 \pm 0.18$ | $88.52 \pm 0.20$ |
| Medium | $78.63 \pm 0.60$ | $88.80 \pm 0.06$ |
| Big | $77.66 \pm 0.33$ | $89.01 \pm 0.09$ |
| **Different $\tau$** | | |
| $\tau = 1$ | 69.89 | 77.13 |
| $\tau = 3$ | 74.52 | 86.60 |
| $\tau = 10$ | 78.31 | 88.93 |
| $\tau = 30$ | 81.15 | 89.14 |
| $\tau = 100$ | 81.79 | 87.66 |
| **Binary Logit Loss** | | |
|  | 65.10 | 70.31 |
| **Fully Connected Last Layer** | | |
| Run 1 | 72.96 | 83.79 |
| Run 2 | 74.77 | 85.18 |
| Run 3 | 74.46 | 85.29 |
| **Codebook-Based Prediction** | | |
| $\tau = 0.1$ | 77.78 | 88.67 |
| $\tau = 0.3$ | 81.57 | 89.16 |
| $\tau = 1$ | 82.21 | 88.26 |
| $\tau = 3$ | 80.35 | 84.71 |
| **Group-Sum Dropout** | | |
| $p = 0.1$ | 79.27 | 88.62 |
| $p = 0.3$ | 81.62 | 88.97 |
| $p = 0.5$ | 81.69 | 88.91 |
| $p = 0.7$ | 82.07 | 88.56 |
| $p = 0.9$ | 81.51 | 86.47 |
| **Convolutional Difflogic** | | |
| $\tau = 1$ | 38.07 | – |
| $\tau = 3$ | 61.86 | – |
| $\tau = 10$ | 80.99 | – |
| $\tau = 30$ | 86.44 | – |
| $\tau = 100$ | 87.41 | – |

Table 7: Test Accuracy (%) Comparison of Models on Q-MNIST.

| | Input | |
|---|---|---|
| | **Discrete** | **Continuous** |
| **Baseline** | | |
| | $97.52 \pm 0.09$ | $97.93 \pm 0.06$ |
| **MLP Baseline** | | |
| Small | $96.45 \pm 0.09$ | $97.60 \pm 0.06$ |
| Medium | $96.67 \pm 0.03$ | $97.79 \pm 0.02$ |
| Big | $97.09 \pm 0.07$ | $97.91 \pm 0.02$ |
| **Different $\tau$** | | |
| $\tau = 1$ | 91.06 | 94.74 |
| $\tau = 3$ | 94.06 | 96.10 |
| $\tau = 10$ | 97.56 | 97.97 |
| $\tau = 30$ | 97.91 | 98.17 |
| $\tau = 100$ | 97.41 | 97.55 |
| **Binary Logit Loss** | | |
| | 74.94 | 74.54 |
| **Fully Connected Last Layer** | | |
| Run 1 | 95.07 | 96.86 |
| Run 2 | 94.65 | 96.91 |
| Run 3 | 95.89 | 97.30 |
| **Codebook-Based Prediction** | | |
| $\tau = 0.1$ | 97.46 | 97.99 |
| $\tau = 0.3$ | 97.88 | 98.19 |
| $\tau = 1$ | 97.65 | 97.81 |
| $\tau = 3$ | 94.59 | 94.64 |
| **Group-Sum Dropout** | | |
| $p = 0.1$ | 97.46 | 97.95 |
| $p = 0.3$ | 97.63 | 98.01 |
| $p = 0.5$ | 97.78 | 98.11 |
| $p = 0.7$ | 97.80 | 98.08 |
| $p = 0.9$ | 96.91 | 97.04 |
| **Convolutional Difflogic** | | |
| $\tau = 1$ | 89.88 | – |
| $\tau = 3$ | 90.20 | – |
| $\tau = 10$ | 97.14 | – |
| $\tau = 30$ | 98.48 | – |
| $\tau = 100$ | 98.74 | – |

Table 8: Test Accuracy (%) Comparison of Models on K-MNIST.

| | Input | |
|---|---|---|
| | **Discrete** | **Continuous** |
| **Baseline** | | |
| | $95.12 \pm 0.14$ | $96.10 \pm 0.08$ |
| **MLP Baseline** | | |
| Small | $92.23 \pm 0.12$ | $94.91 \pm 0.09$ |
| Medium | $92.88 \pm 0.02$ | $95.42 \pm 0.07$ |
| Big | $93.86 \pm 0.21$ | $95.90 \pm 0.24$ |
| **Different $\tau$** | | |
| $\tau = 1$ | 83.83 | 90.57 |
| $\tau = 3$ | 91.63 | 94.33 |
| $\tau = 10$ | 95.00 | 96.01 |
| $\tau = 30$ | 95.88 | 96.44 |
| $\tau = 100$ | 94.32 | 94.70 |
| **Binary Logit Loss** | | |
| | 68.78 | 69.11 |
| **Fully Connected Last Layer** | | |
| Run 1 | 91.29 | 94.83 |
| Run 2 | 91.03 | 94.78 |
| Run 3 | 91.26 | 94.97 |
| **Codebook-Based Prediction** | | |
| $\tau = 0.1$ | 94.73 | 95.90 |
| $\tau = 0.3$ | 95.93 | 96.56 |
| $\tau = 1$ | 94.88 | 95.32 |
| $\tau = 3$ | 88.71 | 88.76 |
| **Group-Sum Dropout** | | |
| $p = 0.1$ | 95.17 | 96.29 |
| $p = 0.3$ | 95.54 | 96.35 |
| $p = 0.5$ | 95.82 | 96.42 |
| $p = 0.7$ | 95.86 | 96.13 |
| $p = 0.9$ | 93.29 | 93.68 |
| **Convolutional Difflogic** | | |
| $\tau = 1$ | 90.31 | – |
| $\tau = 3$ | 88.86 | – |
| $\tau = 10$ | 95.48 | – |
| $\tau = 30$ | 97.58 | – |
| $\tau = 100$ | 97.51 | – |

Table 9: Test Accuracy (%) Comparison of Models on E-MNIST-Letters.

| | Input | |
| --- | --- | --- |
| | **Discrete** | **Continuous** |
| **Baseline** | | |
| | $87.46 \pm 0.14$ | $90.72 \pm 0.14$ |
| **MLP Baseline** | | |
| Small | $86.24 \pm 0.07$ | $90.37 \pm 0.11$ |
| Medium | $87.08 \pm 0.08$ | $90.83 \pm 0.09$ |
| Big | $87.68 \pm 0.17$ | $90.97 \pm 0.09$ |
| **Different $\tau$** | | |
| $\tau = 1$ | 55.68 | 74.34 |
| $\tau = 3$ | 79.63 | 87.74 |
| $\tau = 10$ | 87.62 | 90.66 |
| $\tau = 30$ | 88.85 | 90.15 |
| $\tau = 100$ | 79.04 | 79.32 |
| **Binary Logit Loss** | | |
| | 53.40 | 54.57 |
| **Fully Connected Last Layer** | | |
| Run 1 | 67.33 | 78.65 |
| Run 2 | 69.61 | 80.60 |
| Run 3 | 75.11 | 81.87 |
| **Codebook-Based Prediction** | | |
| $\tau = 0.1$ | 82.90 | 88.97 |
| $\tau = 0.3$ | 88.84 | 90.88 |
| $\tau = 1$ | 86.30 | 86.87 |
| $\tau = 3$ | 72.60 | 73.01 |
| $\tau = 10$ | 58.62 | 59.32 |
| **Group-Sum Dropout** | | |
| $p = 0.1$ | 88.50 | 91.24 |
| $p = 0.3$ | 89.18 | 91.05 |
| $p = 0.5$ | 89.18 | 90.48 |
| $p = 0.7$ | 87.97 | 88.86 |
| $p = 0.9$ | 79.22 | 79.65 |
| **Convolutional Difflogic** | | |
| $\tau = 1$ | 51.15 | – |
| $\tau = 3$ | 86.42 | – |
| $\tau = 10$ | 92.18 | – |
| $\tau = 30$ | 92.69 | – |
| $\tau = 100$ | 91.73 | – |

Table 10: Test Accuracy (%) Comparison of Models on E-MNIST-Balanced.

| | Input | |
| --- | --- | --- |
| | **Discrete** | **Continuous** |
| **Baseline** | | |
| | $80.30 \pm 0.03$ | $83.75 \pm 0.02$ |
| **MLP Baseline** | | |
| Small | $79.25 \pm 0.10$ | $84.04 \pm 0.03$ |
| Medium | $80.13 \pm 0.17$ | $84.52 \pm 0.10$ |
| Big | $80.85 \pm 0.01$ | $84.76 \pm 0.15$ |
| **Different $\tau$** | | |
| $\tau = 1$ | 51.56 | 70.31 |
| $\tau = 3$ | 71.87 | 80.69 |
| $\tau = 10$ | 80.31 | 83.77 |
| $\tau = 30$ | 79.84 | 80.66 |
| $\tau = 100$ | 64.29 | 64.79 |
| **Binary Logit Loss** | | |
| | 48.76 | 49.25 |
| **Fully Connected Last Layer** | | |
| Run 1 | 49.38 | 62.19 |
| Run 2 | 55.19 | 64.29 |
| Run 3 | 67.75 | 75.88 |
| **Codebook-Based Prediction** | | |
| $\tau = 0.1$ | 71.84 | 80.49 |
| $\tau = 0.3$ | 81.20 | 83.81 |
| $\tau = 1$ | 77.74 | 78.20 |
| $\tau = 3$ | 63.07 | 63.30 |
| $\tau = 10$ | 51.41 | 51.93 |
| **Group-Sum Dropout** | | |
| $p = 0.1$ | 81.37 | 84.04 |
| $p = 0.3$ | 81.71 | 83.59 |
| $p = 0.5$ | 80.96 | 82.15 |
| $p = 0.7$ | 78.59 | 79.22 |
| $p = 0.9$ | 65.94 | 66.58 |
| **Convolutional Difflogic** | | |
| $\tau = 1$ | 55.67 | – |
| $\tau = 3$ | 81.02 | – |
| $\tau = 10$ | 85.56 | – |
| $\tau = 30$ | 86.25 | – |
| $\tau = 100$ | 83.32 | – |

Table 11: Test Accuracy (%) Comparison of Models on CIFAR10.

| | Input | |
|---|---|---|
| | **Discrete** | **Continuous** |
| **Baseline** | | |
| | $50.88 \pm 0.87$ | – |
| **MLP Baseline** | | |
| Small | $48.43 \pm 0.15$ | – |
| Medium | $49.33 \pm 0.24$ | – |
| Big | $49.87 \pm 0.48$ | – |
| **Different $\tau$** | | |
| $\tau = 1$ | 37.82 | – |
| $\tau = 3$ | 45.27 | – |
| $\tau = 10$ | 49.88 | – |
| $\tau = 30$ | 53.56 | – |
| $\tau = 100$ | 54.72 | – |
| **Binary Logit Loss** | | |
| | 31.06 | – |
| **Fully Connected Last Layer** | | |
| Run 1 | 41.02 | – |
| Run 2 | 41.25 | – |
| Run 3 | 44.13 | – |
| **Codebook-Based Prediction** | | |
| $\tau = 0.1$ | 49.68 | – |
| $\tau = 0.3$ | 54.33 | – |
| $\tau = 1$ | 55.56 | – |
| $\tau = 3$ | 51.13 | – |
| **Group-Sum Dropout** | | |
| $p = 0.1$ | 51.05 | – |
| $p = 0.3$ | 51.99 | – |
| $p = 0.5$ | 52.95 | – |
| $p = 0.7$ | 53.03 | – |
| $p = 0.9$ | 52.71 | – |
| **Convolutional Difflogic** | | |
| $\tau = 1$ | 27.80 | – |
| $\tau = 3$ | 43.37 | – |
| $\tau = 10$ | 62.22 | – |
| $\tau = 30$ | 65.23 | – |
| $\tau = 100$ | 65.21 | – |

Table 12: Test Accuracy (%) Comparison of Models on CIFAR100.

| | Input | |
|---|---|---|
| | **Discrete** | **Continuous** |
| **Baseline** | | |
| | $22.54 \pm 0.26$ | – |
| **MLP Baseline** | | |
| Small | $18.55 \pm 0.13$ | – |
| Medium | $20.89 \pm 0.11$ | – |
| Big | $22.77 \pm 0.15$ | – |
| **Different $\tau$** | | |
| $\tau = 1$ | 11.40 | – |
| $\tau = 3$ | 17.48 | – |
| $\tau = 10$ | 22.27 | – |
| $\tau = 30$ | 22.89 | – |
| $\tau = 100$ | 17.14 | – |
| **Binary Logit Loss** | | |
| | 9.50 | – |
| **Fully Connected Last Layer** | | |
| Run 1 | 8.92 | – |
| Run 2 | 10.93 | – |
| Run 3 | 9.74 | – |
| **Codebook-Based Prediction** | | |
| $\tau = 0.1$ | 16.86 | – |
| $\tau = 0.3$ | 21.87 | – |
| $\tau = 1$ | 23.12 | – |
| $\tau = 3$ | 17.73 | – |
| **Group-Sum Dropout** | | |
| $p = 0.1$ | 23.50 | – |
| $p = 0.3$ | 23.87 | – |
| $p = 0.5$ | 24.35 | – |
| $p = 0.7$ | 22.75 | – |
| $p = 0.9$ | 17.52 | – |
| **Convolutional Difflogic** | | |
| $\tau = 1$ | 15.88 | – |
| $\tau = 3$ | 25.95 | – |
| $\tau = 10$ | 30.96 | – |
| $\tau = 30$ | 30.67 | – |
| $\tau = 100$ | 25.60 | – |

