# OpenReview forum: "From MNIST to ImageNet: Understanding the Scalability Boundaries of Differentiable Logic Gate Networks"
_ICLR.cc/2026/Conference — ICLR 2026 Conference Withdrawn Submission_

### Official Review · Reviewer_xEip · 2025-10-20

**Soundness:** 2
**Presentation:** 3
**Contribution:** 2
**Rating:** 4
**Confidence:** 4

**Summary:**

This paper investigates the scalability limits of Differentiable Logic Gate Networks (DLGNs) — neural architectures that approximate Boolean logic operations (AND, OR, XOR) in a differentiable manner. The authors systematically analyze whether such models, previously tested only on small binary datasets like MNIST, can extend to complex, high-class-count datasets such as ImageNet-32.
The core contribution lies in evaluating the Group-Sum output layer, which aggregates multiple output neurons per class to improve stability and expressivity. The study reveals that a single hyperparameter — the temperature (τ) — critically governs both optimization behavior and network utilization. Extensive experiments are conducted on synthetic datasets, multi-class MNIST variants, and ImageNet-32, exploring τ’s effect on redundancy, generalization, and pruning dynamics.
The results indicate that while DLGNs match or exceed standard MLPs on structured, low-entropy data, they struggle to generalize on complex visual data. The findings suggest that scalability is not limited by depth or width but by representational rigidity inherent in differentiable logic operations.

**Strengths:**

1.  Comprehensive empirical study: The first work to push DLGNs from simple benchmarks to 1000-class ImageNet-scale datasets, with detailed ablations and sensitivity analyses on τ.

2. Clarity of presentation: The paper is well-written and carefully structured, with transparent methodology and reproducible setup.
Insightful diagnostic findings: Identifies τ as a hidden bottleneck connecting numerical stability, logic-gate expressivity, and neuron redundancy.

3. Hardware awareness: Emphasizes that logic-based differentiable networks retain compatibility with CPU/FPGAs, supporting energy-efficient inference.

4. Bridges symbolic and continuous paradigms: Offers an empirical reference point in the neuro-symbolic spectrum, demonstrating both the promise and pitfalls of logic-based differentiability.

**Weaknesses:**

1. Lack of architectural innovation: The Group-Sum design and τ-tuning dominate the contribution. No fundamentally new structure is proposed to overcome observed scalability bottlenecks.

2. Limited theoretical interpretation: The results remain empirical; there is little formal analysis explaining why τ interacts with class count or neuron redundancy.

3. Missing state-of-the-art comparisons: The study omits benchmarks against binary/quantized neural networks (e.g., BiRealNet, XNOR-Net) or modern symbolic-neural hybrids (e.g., NAIL, DeepProbLog), which are natural competitors in this space.

4. Synthetic dataset simplicity: The binary synthetic benchmarks are too clean and deterministic, offering limited insight into generalization under noise or ambiguity.

5.Underwhelming ImageNet generalization: Sharp accuracy decline beyond 100-class setups weakens the claim that logic networks “scale” in any meaningful sense.

**Questions:**

Detailed Analyses:

This paper illustrates the enduring tension between interpretability and scalability in machine learning.
 DLGNs operate where symbolic reasoning meets continuous optimization — a liminal space where clarity is achieved at the expense of flexibility.

What the authors truly uncover is not a failure of logic, but the need for compositional context. Pure differentiable gates, without hierarchical structure or data-driven feature abstraction, saturate quickly. The study’s most profound insight is that logic without representation depth becomes brittle, while continuous learning without structure becomes opaque.

Thus, even though the paper falls short in performance, it succeeds conceptually: it delineates the boundary where interpretable architectures lose footing against deep models, providing a valuable empirical baseline for hybrid neuro-symbolic research.

The paper presents careful experimentation and honest analysis but does not cross the threshold of conceptual or methodological novelty required for a major venue.

 It would benefit from stronger theoretical grounding, inclusion of modern baselines, and clearer implications for hybrid architectures.
Nevertheless, this is a thoughtful, technically sound, and genuinely valuable study — one that enriches the empirical understanding of differentiable logic models and could serve as a foundation for future neuro-symbolic frameworks integrating logic with learned features.

I expect the authors to defend or rebut the points in the weakness section during the rebuttal phase.

---

### Official Review · Reviewer_DHsT · 2025-10-30

**Soundness:** 1
**Presentation:** 3
**Contribution:** 1
**Rating:** 0
**Confidence:** 3

**Summary:**

The paper investigates the scalability of Differentiable Logic Gate Networks (DLGNs) — networks that implement learnable combinations of logic gates — for large multi-class classification tasks. While most prior work tested DLGNs only on small datasets (e.g., up to 10 classes), this study focuses on large-scale settings (up to ~2000 classes), analyzing output‐layer design (especially the “Group-Sum” layer), the effect of temperature tuning, and the expressiveness and scalability of DLGNs in both synthetic and real‐world classification scenarios.

**Strengths:**

- This is the first work to systematically evaluate DLGNs beyond small 10-class problems, scaling them to tasks with thousands of classes.
- The paper conducts systematic experiments exploring how the choice of output layer (e.g., Group-Sum versus alternatives) and the temperature parameter influence performance in large-class scenarios. This level of detail helps practitioners understand how design choices in DLGNs impact scalability and performance.

**Weaknesses:**

- The technical contributions are quite limited as the paper mainly consists of empirical investigations.
- The primary large-scale, real-world benchmark, ImageNet-32, was a failure. The DLGN models could not match the performance of the baseline MLP.
- The paper hypothesizes that the ImageNet-32 failure is due to the DLGN backbone's small "receptive field" (seeing only ~0.7% of the 9216-dim input). This is a plausible but unproven explanation; the failure could also be due to the logic-based architecture's fundamental difficulty with complex, noisy RGB data, which is a more significant limitation.
- The comparison baseline is a standard MLP. While a fair comparison for the feed-forward DLGN, this is a very weak baseline for image classification. The paper also tests a Convolutional DLGN (CLGN) but does not compare it against a standard CNN, making it hard to gauge its true performance.
- The paper focuses on the scalability of the output layer but notes that DLGNs take a "long time to train". The baseline DLGN uses 64,000 neurons per layer, which is massive. The paper does not address the practical scalability of training these models in terms of time or memory.

**Questions:**

- How would DLGNs perform on non‐classification tasks (e.g., sequence modelling, detection)?
- Are there particular classes of problems (e.g., highly imbalanced classes, incremental class addition) where DLGNs fail or underperform?

---

### Official Review · Reviewer_WVQ8 · 2025-10-31

**Soundness:** 2
**Presentation:** 2
**Contribution:** 1
**Rating:** 2
**Confidence:** 3

**Summary:**

This paper studies the property of Differentiable Logic Gate Networks (DLGNs) on multi-class datasets. Specifically, it finds out that
1. the tempreture parameter tau  in softmax operation of the group soum layer plays an important role in the network's performance: for problems with small number of classes, a large tau is better; and for datasets with large number of classes, a smaller tau is preferred.
2. DLGN performs competitively to MLP in simple classification problems with few classes
Experimental results on synthetic and real world rgb datasets positively supports the claims, and show that for complex real world datasets like imagenet, DLGN performs worse than MLP due to its limited expressivity.

**Strengths:**

1. The paper is well motivated, as studying the scalability of DLGN for large datasets with more classes is important in extending their applicability to complex datasets.
2. The dataset design is reasonable. Specifically, the paper adopts control variable method, to use a synthetic dataset with simple input content, and only varying the number of classes, to study the effect of number of classes alone on the DLGN.
3. The experimental results demonstrate that the choice of tau indeed play an important role in DLGN's performance, and the paper explains the general principle of the selection of tau under different number of classes.

**Weaknesses:**

1. The main finding of the paper is about the role of tau in determining the performance of the DLGN. I'm not sure whether this is sufficient in terms of novelty and contribution, as how tau would affect the activation of logit layer is not new knowledge, and this paper doesn't bring any deeper insights than this.
2. When comparing with MLP, the paper doesn't explain why the three MLP sizes are comparable to the DLGNs, i.e., whether they have similar FLOPs or number of parameters, if this is not shown, it's not clear whether the comparison is fair.

**Questions:**

1. FLOPs and parameter number comparison between the DLGNs used in the experiment and the baseline MLPs should be provided
2. Table 2 shows that convolutional differentiable logic gate networks achieve better performance than DLGNs, the paper should conduct analysis about why convolutional design improve DLGN's performance

---

### Official Review · Reviewer_mMt3 · 2025-10-31

**Soundness:** 2
**Presentation:** 2
**Contribution:** 1
**Rating:** 2
**Confidence:** 4

**Summary:**

The aim of the paper is to analyze and explore the scalability of Differentiable Logic Gate Networks (DLGNs) representing neural models built on differentiable Boolean operations. The paper explores current limitations of the analysis of DLGNs —where the applicability was mostly limited to small-scale datasets like MNIST and CIFAR-10—by evaluating DLGNs on large datasets comprising up to 2000 classes.
In particular the contribution concerns the empirical evaluation of the Group-Sum output layer and of the temperature parameter τ in neuron activation and accuracy in large-class settings.

**Strengths:**

- The paper offers an experimental exploration of DLGN scalability and helps identify key aspects in current architectures.
- It systematically studies the temperature parameter τ, exploring its effect on class separation and neuron utilization.
- The paper includes of both synthetic and real datasets for the analysis

**Weaknesses:**

The main critical point is that the novelty of the contribution is very limited. The paper mainly explores variations of the output layer and temperature scaling but does not introduce fundamentally new architectures beyond the Group-Sum layer. The statement that the work “provides the first large-scale evaluation of DLGNs on datasets with thousands of classes” is technically correct, but the contribution is primarily experimental, not conceptual or methodological.

Figure 1 asserts that DLGNs consistently outperform MLPs across datasets with up to 2000 classes, but unfortunately the text does not specify which datasets or training configurations underlie this statement, hence preventing the reader to gain a critical evaluation of what proposed.

The core technical element seems to reside in Equation (3), where classes are represented by groups of neurons summed as softmax. However, the definition of the grouping parameter k (neurons per class) and the role of τ should be discussed more formally. The current explanation remains empirical and lacks a theoretical rationale.

The paper mentioned efficiency as a primary advantage of what proposed but it's not clear to me whether current hardware is able to support it. One crucial aspects is that FPGA is mentioned but it's not clear to this reviewer whether current hardware is able to support both the training and inference-

The preprocessing pipeline (Section 4.2) appears to play a crucial role. It would be valuable to quantify how much accuracy depends on preprocessing rather than the network architecture itself.

The comparison with baselines is limited since the MLP configuration considered (three layers of 512 neurons with batch normalization) may not represent a strong and general baseline. This makes it difficult to generalize conclusions about “DLGNs vs. MLPs”. Moreover no convolutional or hybrid feed-forward networks are used for comparison.

Minor typographical errors (“Temperatur” in Section 5, for instance) should be corrected, and some figures would benefit from clearer legends.

**Questions:**

- On which specific dataset was Figure 1 based?
- Are DLGNs implemented in FPGA both for inference and for training?
- How critical is the preprocessing step in Section 4.2 for model performance?
- How is k (the number of neurons per class) chosen, and how does it interact with τ?
- Could other output aggregation mechanisms (beyond Group-Sum or temperature scaling) be explored to improve expressiveness?
- Would results differ if stronger or convolutional baselines were used?

---

### Note · Authors · 2025-11-12

I have read and agree with the venue's withdrawal policy on behalf of myself and my co-authors.